# INTRABENCH:
## INTERACTIVE RADIOLOGICAL BENCHMARK

## ABSTRACT

Current interactive segmentation approaches, inspired by the success of META's Segment Anything model, have achieved notable advancements, however they come with substantial limitations that hinder their practical application in real clinical scenarios. These include unrealistic human interaction requirements, such as slice-by-slice operations for 2D models on 3D data, a lack of iterative refinement, and insufficient evaluation experiments. These shortcomings prevent accurate assessment of model performance and lead to inconsistent outcomes across studies.

`IntRaBench` overcomes these challenges by offering a comprehensive and reproducible framework for evaluating interactive segmentation methods in realistic, clinically relevant scenarios. It includes diverse datasets, target structures, and segmentation models, and provides a flexible codebase that allows seamless integration of new models and prompting strategies. Additionally, we introduce advanced techniques to minimize clinician interaction, ensuring fair comparisons between 2D and 3D models. By open-sourcing IntRaBench, we invite the research community to integrate their models and prompting techniques, ensuring continuous and transparent evaluation of interactive segmentation models in 3D medical imaging.

## 1 INTRODUCTION

Accurate segmentation of anatomical structures or pathological areas is crucial in fields like radiology, oncology, and surgery to isolate affected regions, monitor disease progression, treatment planning and guide therapeutic procedures. Traditional supervised medical segmentation models have demonstrated strong performance across a range of anatomies and pathologies (Isensee et al., 2020; 2023; Huang et al., 2023; Ulrich et al., 2023). However, their effectiveness remains heavily constrained by the amount and diversity of available training data, with the quality of human label annotations serving as a critical limiting factor. Consequently, fully autonomous AI solutions have not yet reached performance needed for widespread autonomous clinical applications.

On the other hand, numerous semi-automatic segmentation techniques, not reliant on AI, are already in clinical practice to expedite manual annotation processes Hemalatha et al. (2018). These current ad hoc methods do not tap into the potential of AI-based automation to drastically reduce annotation time. A method that allows clinicians to segment any target with just a single click within the image could greatly enhance the efficiency of clinical workflows.

The release of META's Segment Anything (SAM) model represents a big leap towards making this potential a reality (Kirillov et al., 2023). "SAM" is designed to segment any target through different user interaction methods, including point-based and bounding box prompts. This allows users to easily specify the area of interest by clicking on it or drawing a bounding box around it, making the segmentation process both flexible and intuitive. A particularly powerful feature is the ability for users to iteratively refine initial predictions by adding more positive or negative prompts.

This advanced functionality, in contrast to traditional supervised segmentation methods, has attracted a lot of attention in the medical domain, and led to many studies evaluating and adapting SAM for 3D medical image segmentation (Roy et al., 2023; Deng et al., 2023; Hu et al., 2023; Zhou et al., 2023; Mohapatra et al., 2023; Cheng et al., 2023; Ma et al., 2024; Gong et al., 2023). Moreover, several researchers have been inspired by SAM's capabilities to develop their own methods,

Figure 1: a) Current approaches require clinicians to interact with radiological images slice by slice, leading to increased workload. b) Some models operate natively in 3D and enable full 3D interaction. Only models that accept mask prompts allow iterative refinement of initial predictions with human guidance.

often specifically designed for the 3D nature of radiological data (Du et al., 2024; He et al., 2024; Li et al., 2024; Wang et al., 2024).

Although these domain-specific adaptations on medical data have shown promising progress, many published methods are plagued by pitfalls which obfuscate the efficacy of the models and prevent clinicians and researchers from determining the best methods for their use-cases:

**Applying interactive 2D models to 3D data on a slice-by-slice basis (P1):** Assuming clinicians will interact with each slice individually is unrealistic and undermines the efficiency improvements these methods aim for. Moreover, a slice-by-slice approach introduces an unfair bias when comparing 2D and 3D models, as 3D models typically require only a few interaction per image, leading to significantly fewer interactions and less supervision Cheng et al. (2023); Ma et al. (2024); Zhang & Liu (2023); Wu et al. (2024); Wong et al. (2024).

**Neglecting refinement (P2)**: Many studies assess interactive segmentation methods based on a single interaction step, overlooking the inherent ambiguities in radiological images (Ma et al., 2024; Du et al., 2024; Gong et al., 2023; Bui et al., 2024). Often, a second interaction may be necessary to specify which specific substructure the clinician wants to segment. This could be, e.g. a vessel within the liver, or the necrosis within a tumor, as exemplified in the well-known BraTs segmentation challenge (de Verdier et al., 2024). Furthermore, clinicians often want to adapt the segmentations to their clinic's local protocol or refine them particularly for targets with high inter-rater variability, like pathological structures (Fu et al., 2014; Benchoufi et al., 2020; Hesamian et al., 2019). Overall, there is a notable lack of research exploring realistic, iterative interaction methods for 2D models applied to 3D volumes.

**Obfuscated and insufficient evaluation (P3):** With promptable models only recently garnering great attention, there is a lack of a standardised approach to evaluation, which has led to disparate and incomparable methods, which are at times even obfuscated or insufficient. (i) Not specifying whether predictions were interactively refined or based on a single prompt with multiple points (Cheng et al., 2023; Wang et al., 2024). (ii) Being intransparent on the number of initial prompts given (Du et al., 2024). (iii) Using the best mask rather than the final one after interactive refinement (Wang et al., 2024). (iv) Evaluating predictions slice by slice or on sub-patches of a 3D volume instead of the full image (Roy et al., 2023; Ma et al., 2024; Cheng et al., 2023; He et al., 2024; Li et al., 2024). (v) Excluding targets considered 'too small' neglecting valid targets such as small lesions that are neither tested nor trained on Ma et al. (2024); Cheng et al. (2023); Wang et al.

Figure 2: `IntRaBench` overview. Although our evaluation is performed on entire 3D volumes, the benchmark accommodates both 3D and 2D interactive segmentation methods. While 3D model prompting is relatively straightforward, we introduce prompting and refinement strategies for 2D models that minimize the effort required from human interaction. The benchmark is designed to be extensible, and researchers are encouraged to propose and integrate additional methods seamlessly using our codebase particularly for areas marked by three dots.

(2024). (vi) Many studies only compare against non-promptable models and SAM, rather than any other promptable models trained on medical data (Cheng et al., 2023; Ma et al., 2024; Gong et al., 2023; He et al., 2024). (vii) Lastly, there is an overemphasis on segmenting healthy structures, such as organs, where existing public models already perform well (Wasserthal et al., 2023; Ulrich et al., 2023), instead of focusing on pathologies, where interactive refinement could provide the greatest benefits (Wang et al., 2024; Zhang & Liu, 2023).

To address these pitfalls, we introduce IntRaBench, a reproducible and extendable Interactive Radiological Benchmark. Through it we highlight the most performant 2D and 3D interactive segmentation as well as the best prompting methods in the radiological domain. In this paper, we present experiments carefully designed to replicate a clinical workflow as closely as possible, with the following key contributions:

1. IntRaBench, for the first time, enables a fair comparison of the most influential 2D and 3D interactive segmentation methods. By measuring the number of simulated interactions, a proxy for the "Human Effort", we test different prompting strategies that do not require a slice-wise interaction (P1).

2. We propose effective interaction strategies for refinement of predictions in a 3D volume, without requiring clinicians to interact with each individual slice (P2).

3. We provide a standardized evaluation protocol to generate prompts, select model outputs and compute the segmentation metrics on the entire image across eight datasets, covering various modalities and target structures, including small lesions (P3). Our benchmarking efforts includes a performance comparison against leading interactive segmentation methods in the medical domain.

4. The extendable IntRaBench framework allows developers to a) easily evaluate a new method in a fair manner against established methods and b) easily develop and investigate new prompting strategies.

Through open-sourcing IntRaBench, we invite researchers to integrate their methods into our framework, promoting continuous and equitable assessment that allows to track the overall progress in the field of interactive 3D medical image segmentation reproducibly and transparently.

## 2 INTRABENCH

The Interactive Radiology Benchmark is designed to easily enable a fair and reproducible evaluation of 2D and 3D interactive segmentation methods for 3D radiological image segmentation for the very first time. While prompting 3D models is generally straightforward, we introduce specific prompting

and refinement strategies for 2D models to streamline human interaction and reduce the simulated effort. The proposed benchmark includes seven established models and eight datasets covering different target structures and image modalities. All datasets are public available and we support an automatic download as well as preprocessing for improved usability and reproducibility.

Moreover, the benchmark is built with flexibility in mind, enabling seamless integration of additional methods, as visualized in Fig. 2. Researchers are invited to contribute new approaches, particularly new models, new prompting schemes, and new interesting datasets to the collection. Overall, the design of our benchmark allows for easy testing and validation of novel segmentation methods, making the benchmark a catalyst for advancing methodology for interactive 3D medical image segmentation. In the following we present the different components of `IntRaBench`.

## 2.1 Initial Prompting

Prompts are a key component of any interactive segmentation method and can highly influence overall performance of the underlying method. `IntRaBench`distinguishes between 2 visual prompting types. **Point prompts** correspond to a click of a user in the image, and **box prompts** refer to a box around the target structure. While there is no difference in the action of clicking for 2D and 3D methods, a 3D box requires an additional dimension compared to a 2D box. Notably, some methods also enable a distinction between foreground and background point prompts. While 3D models allow segmenting a 3D volume natively, 2D based models require an interaction for each slice, resulting in excessive effort, which is prohibitive for clinicians as it would take too much time in daily clinical practice. Hence, any meaningful performance comparison must account for this difference in prompting effort.

To increase the feasibility of 2D models for 3D application, it is essential to reduce this effort. We propose two straightforward methods, for both point and box prompts, to explore their performance and provide a proxy for measuring the effort of human interaction.

**Point interpolation**: Let $I \subset N$ be a set of axial indices of all foreground slices. We simulate a user by selecting $n$ foreground points, specifically the center of the largest connected component of slice $i_1, ... i_n \in I$ where the $i_j$ are equally spaced within $I$ and $i_1 = \min(I)$ and $i_n = \max(I)$. Then, we interpolate linearly between each point and the next one and use the intersections of the resulting lines with the axial slices as positive point prompts, as visualized in Fig. 3 c).

**Point propagation**: We simulate a user providing $\min(I)$, $\max(I)$, and a 2D point prompt within the median slice corresponding to the median axial index $i_m$. Given this point, the model generates a segmentation $S_m$ for the median slice. We then select a 'central point,' specifically the center of mass of the largest connected component of $S_m$, to use as a prompt for the slice indexed by $i_{m-1}$. We generate a segmentation $S_{m-1}$ of this slice, and sample from it similarly to generate a prompt for the slice indexed by $i_{m-2}$. This process continues downwards until we segment the slice with the axial index $\min(I)$. The propagation is then repeated upwards, starting from $i_{m+1}$ and using the center of mass of $S_m$, continuing until we segment the slice with the axial index $\max(I)$. This is shown in Fig. 3 e).

**Box interpolation**: We simulate a user providing $n$ 2D bounding boxes, one in each of $i_1, ...., i_n \in I$, the $i_j$ as in point interpolation. Since the boxes are uniquely defined by their minimum and maximum vertices, we can interpolate between the minimum vertices as in point propagation to get a minimum vertex in each axial slice, and similarly get a maximum vertex in each axial slice, this providing a box prompt in each slice. This is shown in Fig. 3 d).

**Box propagation**: We simulate a user providing $\min(I)$, $\max(I)$, and a 2D box prompt within the slice corresponding to the axial index $i_m$, $m$ as in point propagation. The model then generates a segmentation $S_m$ for the median slice. We take a bounding box of $S_m$ and use this as a prompt for the slice indexed by $i_m - 1$. We continue propagating down to $\min(I)$ and then start again from the center and propagate up to $\max(I)$ as in point propagation, but using box prompts instead of point prompts. Fig. 3 f).

While one cares about realistic prompting behavior, `IntRaBench` also supports the previously mentioned slice-wise prompting styles for completeness.

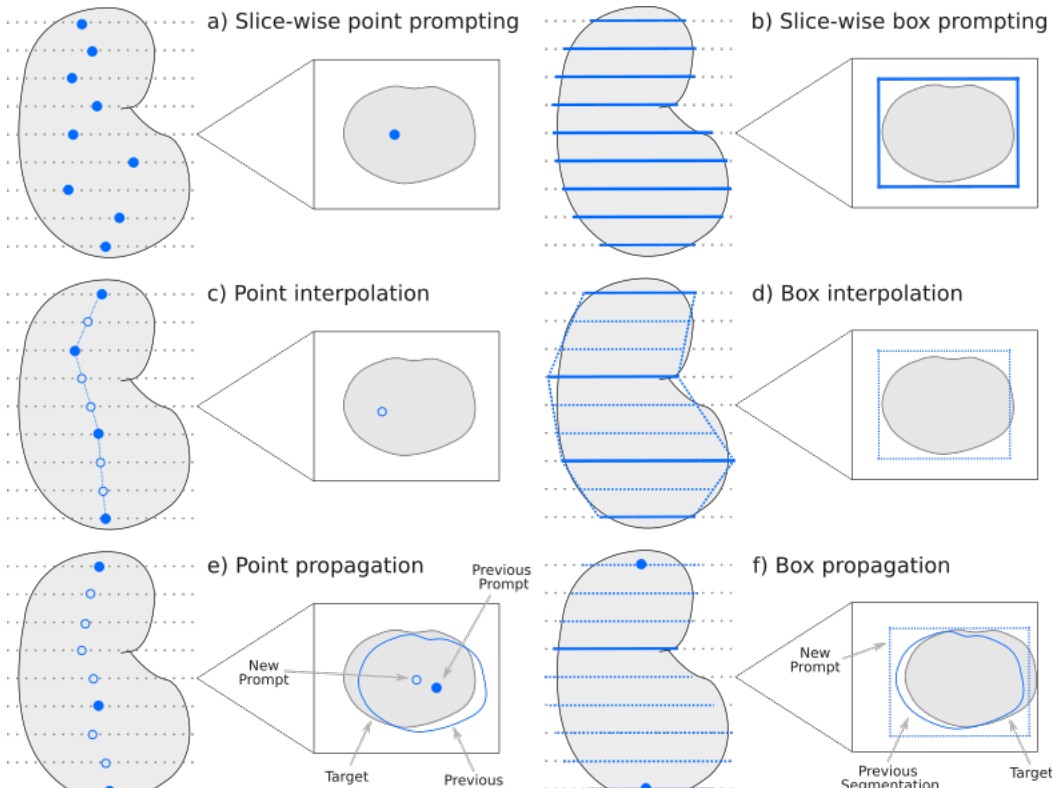

Figure 3: Different promoting schemes for 2D models based on point prompts (on the left) and box prompts (on the right). While a) and b) expect unrealistic human slice-wise interaction,c) and d) illustrate the proposed prompt interpolation schemes, where a human needs to provide prompts for at least 3 slices (4 slices in this case). Prompts for the remaining slices are generated by interpolating between the initial prompts. e) and f) present the proposed prompt propagation methods, where the prompt for each subsequent slice is automatically generated based on the model's prediction from the previous slice. Only the initial slice and upper and lower boundaries require manual prompts.

## 2.2 REFINEMENT PROMPTING

Refinement of previous segmentations is an important aspect of interactive segmentation models, as it allows iteratively improving the segmentation until the desired structure is segmented to a user's demands. Some interactive segmentation models allow for the refinement of initial segmentations by providing the model with the previous prediction along with a new prompt to correct errors, either through foreground clicks on false negative pixels or background clicks on false positive pixels. While this process is straightforward for 3D models, 2D models naively only allow for refinement of one slice at a time, which again places an unrealistic burden on clinicians. Therefore, we present refinement strategies that require a manageable level of effort from the user.

**Scribble refinement:** To represent a user-centric refinement strategy we introduce an algorithm simulating user-created scribble prompts: At each refinement step, our proposed algorithm generates either positive or negative additional prompts. The decision to generate positive prompts follows a Bernoulli trial with success probability $p = n_{fn}/(n_{fn} + n_{fp})$, where $n_{fn}, n_{fp}$ represent the number of false negatives and false positive voxels, respectively.

If positive prompts are selected, we perform a connected component analysis on the false negative voxels. Let $L$ be the largest connected component, we generate a scribble from the bottom to the top of $L$ by taking the centroid of $L$ in each slice to simulate drawing a vertical scribble through the 'middle' of $L$. This simulates a clinician annotating regions that were falsely not segmented.

For 2D models, we then individually feed all slices $i \in I$ where the voxel along the scribble was not predicted, along with the new positive prompt derived from the scribble and the previous prediction

$s \subset S$, back into the model. For 3D models, we feed the whole 3D patch, together with the previous prediction $S$ and all new positive points derived from the scribble into the network in one step.

If negative prompts are selected, we identify a non-axial slice $S_{fp}$ of $S$ that contains the most false positives. Then we generate a contour curve around the ground truth target object at a distance of 2 pixels. We then select a subpart $C$ with a length of 60% of the full curve and sample all pixels $c \in C$ that are false positives to obtain a set of points $D$, simulating a user drawing a few scribbles in areas where the model over-segmented the target. For 2D models, we then generate new slice predictions for each slice containing a point in $D$ by providing the model with the previous prediction as well as new negative prompts: all $d \in D$ which belong to that slice. For 3D models, we again feed the whole 3D patch, together with the previous prediction $S$ and a negative prompt sampled from $D$.

## 2.3 HUMAN EFFORT PROXY

The models' performance is highly dependent on the effort a human puts into initial prompting and refinement of the predicted masks. Generally, the effort required for 3D methods is less than that for 2D methods, although the strategies mentioned above significantly reduce the effort of 2D methods substantially. We aimed to establish a general measure of the effort a method would require from a human user. A more formalized mathematical approach involves assigning degrees of freedom (DoF) to each interaction. For instance, a point corresponds to 3 DoF, a 2D box has 5 DoF (requiring selection of the z-axis and two 2D points), and a 3D box consists of 6 DoF. However, point interpolation has 9 DoF, whereas point propagation only has 5 DoF, since it requires just the axial coordinate rather than both minimum and maximum points with 3 DoF each. From the user's perspective, however, identifying the z-coordinate demands the same level of effort as selecting a 3D coordinate by clicking at the target structure's endpoint along the z-axis. Similarly, an arbitrary scribble has significantly more DoF than a straight or parabolic line, yet the difference in effort for the user is minimal. Therefore, we define user effort in terms of the number of interactions required for a specific task. While not an exact measure, this method offers the most practical estimation of the actual effort involved from the user's perspective.

## 2.4 INTERACTIVE METHODS

In our comprehensive benchmark, we include various interactive segmentation methods. Fig. 1 illustrates the types of prompts each method supports. Iterative refinement is only possible for methods that allow a (previously predicted) mask as a prompt.

**SAM** is the most prominent model from the natural image domain, that inspired many researchers to evaluate and adapt it to the domain of radiological medical images. It was trained on iteratively generated and curated 1B masks and 11M images, but not explicitly on radiological images. META's Segment anything model was the first to popularize interactive segmentation models (Kirillov et al., 2023).

**SAM2** is an extension of SAM that was trained on even more images and introduced support for video data (Ravi et al., 2024).

**MedSAM** is an adaptation of SAM that fine-tuned SAM's weights on 1,570,263 image-mask pairs from the medical domain. It supports only a single forward pass without refinement and is limited to box prompts (Ma et al., 2024).

**SAM-Med 2D** is another adaptation of SAM, fine-tuned on 4.6 million images with 19.7 million masks from the medical domain. Unlike MedSAM, it supports points, boxes, and mask prompts, allowing for refinement (Cheng et al., 2023).

**SAM-Med 3D** incorporates a transformer-based 3D image encoder, 3D prompt encoder, and 3D mask decoder. It was trained from scratch using 22,000 3D images and 143,000 corresponding 3D masks and supports point and mask prompts and also allows for refinement (Wang et al., 2024).

**SAM-Med 3D Turbo** is an updated version of SAM-Med 3D trained on a larger dataset collection of 44 datasets for improved performance. It supports the same prompt styles as SAM-Med 3D (Wang et al., 2024).

**SegVol** is an interactive 3D segmentation model based on a 3D adaptation of a ViT (Dosovitskiy, 2020) that was trained on 96K unlabelled CT images and fine-tuned with 6K labeled CT images. It

Table 1: Overview over all Datasets.

| Dataset | Modality | Targets | Images |
|---|---|---|---|
| D1 MS Lesion (Muslim et al., 2022) | MRI (T2 Flair) | MS Lesions | 60 |
| D2 HanSeg (Podobnik et al., 2023) | MR (T1) | 30 Organs at risk | 42 |
| D3 HNTSRMFG (Wahid et al., 2024) | MRI (T2) | Ropharyngeal cancer & metastatic lymph nodes | 135 |
| D4 RiderLung (Zhao et al., 2015) | CT | Lung lesions | 58 |
| D5 LNQ (Dorent et al., 2024) | CT | Mediastinal lymph nodes | 513 |
| D6 LiverMets (Simpson et al., 2023) | CT | Liver metastases | 171 |
| D7 Adrenal ACC (Moawad et al., 2023) | CT | Adrenal tumors | 53 |
| D8 HCC Tace (Moawad et al., 2021) | CT | Liver, Liver tumors | 65 |

supports points and bounding boxes as spatial prompts but does not allow iterative refinement (Du et al., 2024).

Aside from these models there exist other notable interactive models, such as Vista3D(He et al., 2024), 3D Sam Adapter(Gong et al., 2023) and Prism (Li et al., 2024). However, while being promptable, they are closed-set, i.e. not trained to segment any arbitrary prompted class. Subsequently they were not considered for this benchmark.

## 2.5 DATASETS

Dataset selection was a non-trivial problem for this benchmark: While models that were originally introduced in the natural image domain rarely see any radiological 3D data, the medical counterparts were often trained on all publicly available data that the authors could obtain. For example, MedSAM was trained using more than 60 publicly available datasets (Ma et al., 2024). Although these methods conducted their final validation on excluded datasets or at least on separate test subsets of images, the test data varies between models. As a result, identifying annotated datasets with interesting target structures that were not part of any methods training set has proven challenging.

Nevertheless, we assembled a diverse collection of ten lesser known or recently released public datasets featuring various pathologies and organs, including CT and MRI images. Specific details of these are provided in Table 1. To enhance reproducibility and eliminate barriers of entry for non domain experts, we automated the dataset download and preprocessing, minimizing any required domain knowledge to use the benchmark. However, due to the sparsity of labeled datasets we urge developers to exclude these datasets from their train dataset selection, as this would compromise the integrity of a clean evaluation.

## 2.6 EVALUATION

All interactive segmentation methods identify their target structure based on a spatial prompt, inherently resulting in instance segmentation. As a result, we perform the evaluation on an instance-by-instance basis. Unlike in object detection, each prompt already provides information on the localization of the target structure, hence detection metrics like F1-Score are not relevant, hence we rely solely on the Dice Similarity Coefficient (DSC) score as a metric. The instance-wise DSC metric is then averaged per case (i.e. per image volume), and further aggregated across all cases in the dataset, as recommended by Maier-Hein et al. (2024). For better presentation, we averaged the dice across all classes of a dataset and also specify how many human interactions are simulated.

## 3 EXPERIMENTS

We evaluate all seven models across various initial prompting scenarios under both realistic and unrealistic effort settings. Following this, we conduct interactive experiments to simulate human refinement of model predictions. Due to vast amount of data, we only provide a condensed version of the results for easier insights. Detailed results and the number of human interactions are provided in the Appendix C.

## 3.1 INITIAL PREDICTION

**Unrealistic effort:** As an upper baseline, we begin with an idealized and unrealistic scenario where each slice is prompted individually for all 2D models. In this setting, we evaluate different numbers of point prompts per slice (PPS), as well as alternating positive and negative prompts (± PPS), and slice-wise box prompts with varying numbers of boxes per slice (BPS). Fig. 4 shows that models employing box prompts achieved significantly higher average Dice scores, with SAM2 demonstrating the strongest performance across all models. Conversely, point-based prompts performed poorly, particularly for small target regions, such as small MS lesions in dataset D1 (see Table 2). SAM Med2D outperforms non-medical models for point prompts. Although alternating positive and negative prompts led to improvements, and increasing the number of point prompts yielded some performance gains, these improvements were minimal compared to the marked superiority of box-based prompts. These results highlight the limitations of point prompts, especially in cases involving small or complex anatomical structures, and emphasize the robustness of box prompts in achieving higher segmentation accuracy.

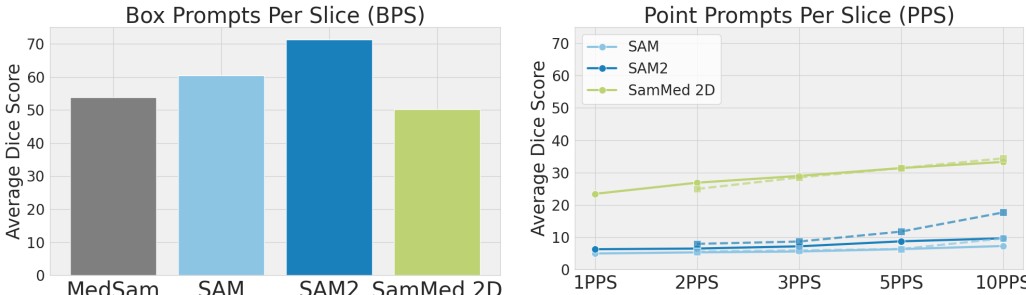

Figure 4: **Unrealistic prompting of 2D Boxes each slice performs best.** When comparing model's prompted with one Box Prompts Per Slice (BPS) (left) with various Point Prompts per Slice (PPS) (right) boxes perform better. While alternating positive and negative points (dashed lines) is slightly superior to only positive points the gap between points and boxes remains large. Denote the different y-axis scalings.

**Realistic Effort:** To simulate a human-in-the-loop scenario, we evaluate various prompting strategies that avoid slice-by-slice interaction. As described in Section 2, for 2D models, we test point and box interpolation, as well as propagation, using different numbers of initial prompts. For 3D models, we explore varying numbers of Point prompts Per Volume (PPV) and 3D box prompts. Fig. 5 presents the following key findings:

1. For all models, box interpolation with 3 or 5 initial 2D boxes is sufficient to achieve results similar to slice-wise box prompting (BPS).

2. For SAMMed 2D, using 3 points with simple point interpolation achieves results comparable to prompting every slice.

3. SAM 2 outperforms specialized medical models across all prompting schemes using box interpolation.

4. Among 3D models, only SegVol is competitive to 2D models that use box prompts.

5. Both box and point propagation perform worse than interpolation, though this may improve as models evolve.

## 3.2 INTERACTIVE REFINEMENT

Finally, we evaluate the performance of the models during iterative refinement. For 2D models, this involves prompting on a slice-by-slice basis. As illustrated in Fig. 6 (left), adding refinement prompts to each slice results in a substantial performance boost. Although the proposed scribble-based refinement consistently improves outcomes, it does not achieve the same level of improvement as adding a prompt to every slice, which is expected since not all slices receive new prompts. We

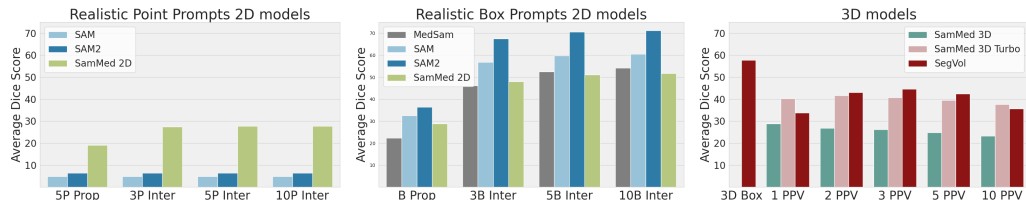

Figure 5: **Simple Interpolation Strategies Match Unrealistic Slice-Wise Prompting.** Sampling prompts from the interpolated connection between three initial prompts yields similar performance for SAM2 as slice-wise prompting (left). This is also observed for box interpolation across all models (middle). 3D models perform worse than 2D methods when only a few points are provided, while SegVol demonstrates that using a 3D box is superior to points (right).

observed that for 2D models, it is crucial to provide the initial prompts during the refinement process. 2D models tend to over segment the target, filling the entire slice foreground. The absence of the initial prompt leads to a complete loss of target location information, as the initial predicted mask is highly inaccurate. Our refinement likely generates negative additional prompts due to the large number of false positive pixels. In Table 5, we present refinement results from initial predictions produced by 3 Box Interpolation. In this case, we did not include the previous point in the iterative prompts, which resulted in a performance decline during refinement.

For 3D models, iterative refinement also led to consistent performance gains. Both randomly sampled prompts and those derived from refinement scribbles improved performance with each refinement iteration. Although SegVol initially performs best without iterative refinement, it lacks support for further refinement. In contrast, SamMed 3D Turbo surpasses SegVol after several refinement steps.

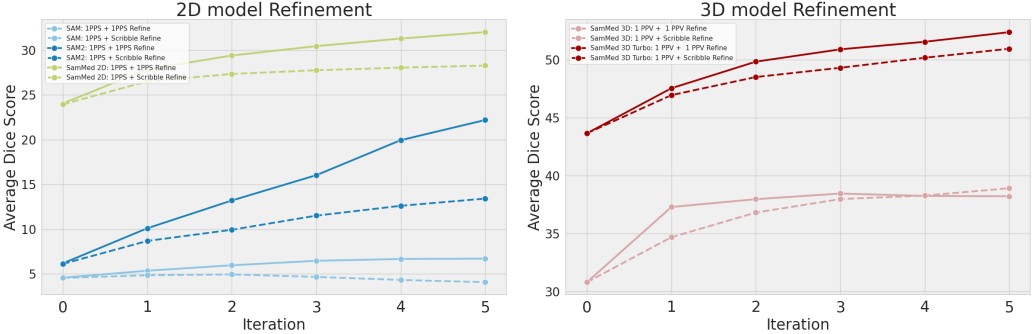

Figure 6: **All models demonstrate significant improvements from iterative refinement**. Results for 2D models are shown on the left, and for 3D models on the right. Dashed lines represent the use of the proposed scribble refinement. While the unrealistic scenario of one refinement point per slice yields better performance, the proposed scribble refinement consistently enhances results across iterations for 2D models.

### 3.3 DISCUSSION AND CONCLUSION

In this paper we introduced `IntRaBench` and with it, compared the performance of 2D and 3D interactive segmentation models in 3D medical imaging. We provide a holistic and transparent overview of the current state-of-the-art and highlight key findings that offer practical insights:

1. **Bounding Boxes Outperform Points:** Bounding boxes consistently outperform point-based inputs by providing better spatial context, which leads to improved segmentation accuracy, especially for complex structures in radiological images. Point-based prompts lack this context, resulting in poorer performance.

2. **Iterative Refinement is Essential:** The ability to iteratively refine segmentations significantly enhances model performance, particularly in challenging cases. Models that allow multiple rounds of corrections show better accuracy, making this feature crucial for clinical applications. For example SegVol reached highest performance in a static setting, however SamMed 3D Turbo is able to exceed SegVol given a few interactions, highlighting the importance of refinement.

3. **Realistic 2D prompting can match unrealistic prompting:** Our introduced realistic prompting styles are able to reach **??** and match unrealistic prompting 2D prompting methods. This unlocks 2D methods for actual clinical workflows without any performance penalties.

4. **Points Underperform Compared to Literature:** Contrary to claims in previous literature, point-based methods underperformed, likely due to previous work training and evaluating their methods on simpler target structures. Previous work mostly focused largely on high-contrast tasks, and excluded small target objects inflating expectations for these methods.

**Implications** `IntRaBench` suggests that bounding boxes and iterative refinement should be prioritized in the design of segmentation models for medical imaging, particularly when addressing complex radiological images. Furthermore, it underscores the importance of including diverse, difficult tasks in training data to improve model generalization for clinical use. It is also crucial to test 2D models in scenarios that simulate real human interaction, ensuring that segmenting a volumetric image does not require unreasonable effort by prompting the model slice by slice.

A key limitation of this work is that it only simulates real clinical settings. While this approach provides valuable insights into model performance providing a proxy for the simulated human effort, it falls short of capturing the full complexity and practical challenges of actual clinical workflows. As future work, a comprehensive study involving clinicians is essential to assess different prompting strategies in real-world environments. Such a study should not only evaluate segmentation performance but also measure the time required for annotation, offering critical insights into the practical feasibility and efficiency of these models in clinical practice.

To conclude, our proposed IntRaBench benchmark presents a powerful tool for the future of interactive segmentation research in medical imaging, serving as a catalyst for innovative solutions by enabling a fair and reproducible comparison between leading methods. One of the standout potentials is its ability to streamline the evaluation of both 2D and 3D segmentation models, allowing for more realistic and clinically relevant testing conditions. By focusing on human interaction and the efficiency of iterative refinement, IntRaBench opens new avenues for research, including understanding the impact of different interaction strategies and how they reduce clinician effort. Not only does this benchmark address the existing gaps in evaluation standardization, but it also offers a unique opportunity to refine segmentation performance on pathologies often overlooked, such as small lesions. The open-source nature of the benchmark further encourages continuous contributions, allowing researchers to test new methods and prompting strategies seamlessly within this framework. Future work using IntRaBench can reveal novel insights into the balance between performance and clinician involvement, fostering advancements that may lead to improved medical workflows. This potential to improve real-world clinical applications, especially by reducing the labor intensity of medical professionals, marks IntRaBench as a crucial tool in catalyzing meaningful research progress.

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

## A  APPENDIX

## B  MODEL SPECIFICATIONS

### B.1  SAM

SAM is compatible with multiple image encoders, particularly the ViT family from Dosovitskiy et al. (2021). We used the default and best-performing model with ViT-Huge. To ensure high-quality inputs for the model, we performed slice-wise inference by slicing through the valong the through-plane. Each slice was normalized by first clipping 0.5th and 99.5th percentile of the volume's intensity distribution and then scaling the values between volume minimum and maximum to [0,1], followed by a rescaling to [0,255]. The image was repeated three times along the channel axis to produce an RGB-like image. The images were resized so their longest side was 1024 pixels, padding the shorter side to 1024 pixels if needed to maintain square dimensions. Finally, the images were again normalized using the model's pre-stored mean and standard deviation as suggested by the original implementation. Inference was restricted to slices containing foreground. After prediction, the slices were reassembled into a volume, inverse transformed to the original coordinate system, and metrics were computed in the native image space.

### B.2  SAM2

SAM2 supports multiple image encoders, specifically the Hiera family of Ryali et al. (2023). We used the best-performing model, Hiera-L. We clip the volumes, slice along the through-plane and make the images RGB-like just as with SAM. The images are then rescaled to $1024 \times 1024$ pixels and again normalized using mean and standard deviation provided together with the pretrained weights. Aggregation and inverse transformation are then performed similarly to SAM.

### B.3  MEDSAM

To apply the model slice-wise, we slice the input volume as with SAM, and then clip each slice based on their 0.5th and 99.5th percentile values. The images are then made RGB-like by repeating thrice along a new channel-dimension, rescaled to $1024 \times 1024$ pixels and then normalised to [0,1]. Aggregation and inverse transformation is performed similarly as with SAM.

### B.4  SAM-MED2D

To apply the model slice-wise, we slice the input volume as with SAM, and then clip each slice based on their 0.5th and 99.5th percentile values same as with MedSAM. The slices are then made RGB-like and converted to a [0,255] scale as in SAM's preprocessing. The slices are then standardised using a mean and standard deviation provided along with the model, and resized to $256 \times 256$ pixels. Aggregation and inverse transformation is performed similarly as with SAM.

### B.5  SAM-MED3D

The model is not applied slice-wise so no reorientation is needed. The volume is respaced to $1.5 \times 1.5 \times 1.5$ mm and then clipped based on its 0.5th and 99.5th percentiles. SAMMed3D performs inference on a 128x128x128 pixel crop, which in a first inference pass we take centered around our point prompt, if there is only one point prompt passed, and around the centroid of our prompts if multiple points are passed simulatenously. For subsequent passes (in the case of iterative refinement), the crop remains unchanged. The predicted crop is inserted back in its correct position within the wider coordinate system, and then respaced back to the original spacing so that evaluation takes place in the coresponding native image space.

### B.6  SAM-MED3D TURBO

SAM-Med3D Turbo is an updated checkpoint for SAMMed-3D and so we perform the same pre- and postprocessing.

### B.7 SEGVOL

SegVol uses all voxels with value exceeding the volume mean to calculate the 0.5th and 99.5th percentile of these voxels. Intensity values are clipped by these percentiles, and the mean and standard deviation of the foreground voxels are used for normalization. The values are then rescaled to a [0,1]. Finally, the volume is cropped to its foreground. A first 'zoom-out' inference is performed on this image, followed by a 'zoom-in' sliding window inference. The predicted volume is then transformed back to the original space and compared with the unprocessed ground truth to calculate metrics.

## C ADDITIONAL RESULTS

### C.1 INITIAL PREDICTION - UNREALISTIC EFFORT

| Prompter | Model | Interactions | D1 | D2 | D3 | D4 | D5 | D6 | D7 | D8 | Average |
|---|---|---|---|---|---|---|---|---|---|---|---|
| 1PPS | SAM | 1X | 0.81 | 7.63 | 3.13 | 2.06 | 1.42 | 1.0 | 7.86 | 12.47 | 4.55 |
| 1PPS | SAM2 | 1X | 1.25 | 7.7 | 4.08 | 3.9 | 3.07 | 1.51 | 9.6 | 16.35 | 5.93 |
| 1PPS | SamMed 2D | 1X | 10.72 | 30.01 | 24.55 | 28.83 | 24.22 | 12.18 | 30.83 | 31.24 | 24.07 |
| 2±PPS | SAM | 2X | 0.95 | 7.8 | 3.41 | 3.29 | 2.12 | 1.06 | 8.89 | 14.31 | 5.23 |
| 2±PPS | SAM2 | 2X | 3.39 | 8.66 | 7.01 | 4.89 | 4.03 | 1.93 | 13.18 | 17.03 | 7.51 |
| 2±PPS | SamMed 2D | 2X | 11.88 | 31.73 | 27.62 | 33.29 | 25.33 | 13.67 | 32.55 | 31.5 | 25.95 |
| 2PPS | SAM | 2X | 0.88 | 7.64 | 3.15 | 2.56 | 1.88 | 1.03 | 8.1 | 14.08 | 4.91 |
| 2PPS | SAM2 | 2X | 1.6 | 7.73 | 4.44 | 3.9 | 3.12 | 1.54 | 9.81 | 16.74 | 6.11 |
| 2PPS | SamMed 2D | 2X | 11.63 | 33.47 | 28.9 | 32.88 | 26.12 | 13.49 | 37.85 | 36.41 | 27.59 |
| 3±PPS | SAM | 3X | 1.02 | 7.87 | 3.6 | 3.61 | 2.65 | 1.18 | 9.66 | 15.07 | 5.58 |
| 3±PPS | SAM2 | 3X | 4.22 | 9.05 | 7.78 | 5.57 | 4.56 | 2.08 | 14.6 | 17.94 | 8.23 |
| 3±PPS | SamMed 2D | 3X | 12.83 | 35.29 | 31.72 | 36.4 | 27.2 | 14.89 | 40.42 | 36.91 | 29.46 |
| 3PPS | SAM | 3X | 0.96 | 7.76 | 3.18 | 2.91 | 2.34 | 1.18 | 8.54 | 14.7 | 5.2 |
| 3PPS | SAM2 | 3X | 2.29 | 8.27 | 5.64 | 4.41 | 3.74 | 1.79 | 10.87 | 17.2 | 6.78 |
| 3PPS | SamMed 2D | 3X | 11.95 | 34.85 | 31.8 | 33.61 | 26.51 | 14.08 | 43.26 | 40.16 | 29.53 |
| 5±PPS | SAM | 5X | 1.01 | 7.74 | 3.98 | 4.31 | 2.28 | 1.31 | 11.76 | 15.97 | 6.04 |
| 5±PPS | SAM2 | 5X | 7.37 | 10.81 | 10.41 | 6.79 | 6.57 | 3.01 | 23.19 | 20.41 | 11.07 |
| 5±PPS | SamMed 2D | 5X | 13.95 | 37.57 | 35.63 | 39.39 | 28.53 | 16.32 | 46.49 | 41.05 | 32.37 |
| 5PPS | SAM | 5X | 1.14 | 8.3 | 4.0 | 3.94 | 2.96 | 1.35 | 10.55 | 15.5 | 5.97 |
| 5PPS | SAM2 | 5X | 4.04 | 9.4 | 7.91 | 5.74 | 4.5 | 2.01 | 15.05 | 17.77 | 8.3 |
| 5PPS | SamMed 2D | 5X | 12.26 | 35.67 | 34.55 | 33.08 | 26.58 | 14.63 | 50.88 | 45.0 | 31.58 |
| 10±PPS | SAM | 10X | 2.03 | 9.69 | 7.13 | 8.59 | 6.32 | 1.68 | 18.79 | 21.29 | 9.44 |
| 10±PPS | SAM2 | 10X | 15.61 | 15.55 | 15.78 | 11.5 | 12.12 | 7.65 | 33.49 | 23.09 | 16.85 |
| 10±PPS | SamMed 2D | 10X | 15.02 | 38.97 | 39.67 | 42.49 | 29.29 | 18.55 | 52.98 | 46.17 | 35.39 |
| 10PPS | SAM | 10X | 1.3 | 8.86 | 5.29 | 5.3 | 3.46 | 1.4 | 13.45 | 16.73 | 6.98 |
| 10PPS | SAM2 | 10X | 4.61 | 9.81 | 7.58 | 6.12 | 4.8 | 2.2 | 19.81 | 18.38 | 9.16 |
| 10PPS | SamMed 2D | 10X | 12.03 | 34.68 | 36.39 | 30.38 | 25.15 | 15.21 | 58.07 | 51.41 | 32.92 |
| Box PS | MedSam | 2X | 40.63 | 50.21 | 55.5 | 60.4 | 45.73 | 46.43 | 67.75 | 70.23 | 54.61 |
| Box PS | SAM | 2X | 13.27 | 60.96 | 68.26 | 63.2 | 66.22 | 74.72 | 70.16 | 69.46 | 60.78 |
| Box PS | SAM2 | 2X | 70.25 | 66.08 | 73.21 | 73.06 | 72.07 | 76.51 | 73.33 | 67.9 | 71.55 |
| Box PS | SamMed 2D | 2X | 28.2 | 48.85 | 57.04 | 64.45 | 46.07 | 45.22 | 62.07 | 63.59 | 51.94 |

Table 2: Experimental results simulating unrealistic effort of a clinician prompting each slice of a 3D volume. 'PPS' and 'BPS' represent points per slice or box per slice, respectively. 'X' implies that each interaction is replicated for every slice, multiplying the clinician's effort across the entire volume.

## C.2    SINGLE FORWARD PASS - REALISTIC EFFORT

| Prompter | Model | Interactions | D1 | D2 | D3 | D4 | D5 | D6 | D7 | D8 | Average |
|----------|-------|--------------|-----|-----|-----|-----|-----|-----|-----|-----|---------|
| 3P Inter | SAM | 3 | 0.84 | 8.45 | 3.49 | 2.22 | 1.67 | 1.03 | 8.28 | 12.67 | 4.83 |
| 3P Inter | SAM2 | 3 | 1.38 | 8.51 | 4.75 | 4.18 | 3.41 | 1.63 | 10.34 | 17.01 | 6.40 |
| 3P Inter | SamMed 2D | 3 | 11.61 | 32.62 | 28.00 | 34.33 | 27.71 | 13.47 | 36.29 | 35.80 | 27.48 |
| 5P Inter | SAM | 5 | 0.84 | 8.45 | 3.50 | 2.17 | 1.70 | 1.02 | 8.29 | 12.61 | 4.82 |
| 5P Inter | SAM2 | 5 | 1.38 | 8.52 | 4.73 | 4.10 | 3.37 | 1.63 | 10.46 | 16.98 | 6.40 |
| 5P Inter | SamMed 2D | 5 | 11.71 | 33.07 | 28.46 | 34.42 | 27.87 | 13.59 | 36.81 | 35.65 | 27.70 |
| 10P Inter | SAM | 10 | 0.84 | 8.46 | 3.48 | 2.21 | 1.71 | 1.03 | 8.26 | 12.64 | 4.83 |
| 10P Inter | SAM2 | 10 | 1.38 | 8.51 | 4.76 | 4.14 | 3.43 | 1.62 | 10.41 | 17.01 | 6.41 |
| 10P Inter | SamMed 2D | 10 | 11.74 | 33.44 | 28.45 | 34.71 | 27.95 | 13.58 | 36.67 | 35.10 | 27.71 |
| 5P Prop | SAM | 7 | 1.09 | 8.31 | 3.60 | 1.78 | 1.70 | 0.99 | 7.84 | 13.70 | 4.88 |
| 5P Prop | SAM2 | 7 | 3.77 | 8.52 | 4.59 | 3.47 | 3.13 | 1.60 | 8.95 | 17.52 | 6.44 |
| 5P Prop | SamMed 2D | 7 | 10.87 | 23.25 | 14.18 | 17.91 | 17.57 | 9.92 | 27.25 | 31.84 | 19.10 |
| B Prop | MedSam | 4 | 2.97 | 24.86 | 22.75 | 22.76 | 23.38 | 23.81 | 28.55 | 29.89 | 22.37 |
| B Prop | SAM | 4 | 0.89 | 31.47 | 37.20 | 37.59 | 36.38 | 38.49 | 43.20 | 35.47 | 32.59 |
| B Prop | SAM2 | 4 | 3.82 | 37.03 | 43.31 | 40.82 | 41.08 | 40.88 | 46.47 | 37.39 | 36.35 |
| B Prop | SamMed 2D | 4 | 2.82 | 27.51 | 30.17 | 36.12 | 24.91 | 25.38 | 38.00 | 45.60 | 28.81 |
| 3B Inter | MedSam | 6 | 40.14 | 43.62 | 46.18 | 48.16 | 41.41 | 40.75 | 54.42 | 54.98 | 46.21 |
| 3B Inter | SAM | 6 | 13.13 | 56.31 | 63.34 | 58.64 | 64.26 | 71.35 | 65.52 | 61.78 | 56.79 |
| 3B Inter | SAM2 | 6 | 69.76 | 59.60 | 67.71 | 69.08 | 69.06 | 73.11 | 68.36 | 63.24 | 67.49 |
| 3B Inter | SamMed 2D | 6 | 27.83 | 45.06 | 52.70 | 60.66 | 44.46 | 42.34 | 55.31 | 55.81 | 48.02 |
| 5B Inter | MedSam | 10 | 40.55 | 47.77 | 52.64 | 57.67 | 44.75 | 45.00 | 64.60 | 66.88 | 52.48 |
| 5B Inter | SAM | 10 | 13.25 | 58.54 | 66.81 | 62.58 | 65.76 | 73.63 | 69.51 | 68.25 | 59.79 |
| 5B Inter | SAM2 | 10 | 70.13 | 63.09 | 71.53 | 72.57 | 71.32 | 75.52 | 72.67 | 67.25 | 70.51 |
| 5B Inter | SamMed 2D | 10 | 28.11 | 47.31 | 56.04 | 64.01 | 45.62 | 44.51 | 60.67 | 62.32 | 51.07 |
| 10B Inter | MedSam | 20 | 40.63 | 49.36 | 54.89 | 60.16 | 45.59 | 46.15 | 67.17 | 69.54 | 54.19 |
| 10B Inter | SAM | 20 | 13.27 | 59.95 | 67.81 | 63.17 | 66.12 | 74.35 | 70.02 | 69.21 | 60.49 |
| 10B Inter | SAM2 | 20 | 70.25 | 64.91 | 72.65 | 73.05 | 71.94 | 76.12 | 73.18 | 67.62 | 71.21 |
| 10B Inter | SamMed 2D | 20 | 28.19 | 48.28 | 56.73 | 64.40 | 46.03 | 45.01 | 61.89 | 63.30 | 51.73 |

Table 3: Experimental results simulating a realistic clinician's effort. 'PPS' and 'PPV' represent points per slice or volume, respectively. 'B Prop' and 'P Prop' denote the introduced box and point propagation schemes, while 'B Inter' and 'P Inter' refer to the introduced box and point interpolation methods.

| Prompter | Model | Interactions | D1 | D2 | D3 | D4 | D5 | D6 | D7 | D8 | Average |
|---|---|---|---|---|---|---|---|---|---|---|---|
| 1 center PPV | SamMed 3D NORM | 1 | 2.04 | 17.00 | 24.06 | 27.16 | 15.09 | 19.64 | 72.67 | 53.29 | 28.87 |
| 1 center PPV | SamMed 3D Turbo NORM | 1 | 5.12 | 31.27 | 46.07 | 34.33 | 15.90 | 46.38 | 82.95 | 59.62 | 40.20 |
| 1 center PPV | SegVol NORM | 1 | 9.95 | 29.15 | 37.68 | 31.36 | 3.17 | 33.50 | 73.11 | 52.07 | 33.75 |
| 2 center PPV | SamMed 3D NORM | 2 | 1.84 | 16.37 | 23.15 | 24.45 | 13.12 | 18.18 | 71.21 | 45.57 | 26.74 |
| 2 center PPV | SamMed 3D Turbo NORM | 2 | 5.27 | 30.54 | 45.71 | 33.26 | 15.74 | 46.76 | 84.84 | 70.67 | 41.60 |
| 2 center PPV | SegVol NORM | 2 | 11.19 | 34.62 | 48.89 | 58.45 | 12.30 | 52.51 | 72.20 | 54.25 | 43.05 |
| 3 center PPV | SamMed 3D NORM | 3 | 1.74 | 16.15 | 22.48 | 23.41 | 12.17 | 17.31 | 70.31 | 46.23 | 26.23 |
| 3 center PPV | SamMed 3D Turbo NORM | 3 | 5.10 | 30.42 | 43.80 | 30.84 | 15.27 | 46.55 | 85.79 | 68.53 | 40.79 |
| 3 center PPV | SegVol NORM | 3 | 11.53 | 34.43 | 48.66 | 57.51 | 19.05 | 53.14 | 71.56 | 61.08 | 44.62 |
| 5 center PPV | SamMed 3D NORM | 5 | 1.70 | 15.82 | 21.86 | 21.43 | 11.19 | 16.04 | 66.49 | 43.95 | 24.81 |
| 5 center PPV | SamMed 3D Turbo NORM | 5 | 5.03 | 29.72 | 42.74 | 26.96 | 14.66 | 46.49 | 85.91 | 64.39 | 39.49 |
| 5 center PPV | SegVol NORM | 5 | 11.69 | 34.66 | 49.40 | 52.47 | 25.32 | 52.97 | 64.02 | 49.05 | 42.45 |
| 10 center PPV | SamMed 3D NORM | 10 | 1.69 | 15.44 | 20.87 | 20.78 | 10.38 | 14.54 | 61.01 | 41.73 | 23.30 |
| 10 center PPV | SamMed 3D Turbo NORM | 10 | 5.00 | 28.89 | 40.04 | 21.51 | 13.09 | 46.03 | 86.05 | 61.00 | 37.70 |
| 10 center PPV | SegVol NORM | 10 | 11.69 | 33.70 | 45.49 | 47.32 | 26.67 | 51.55 | 41.98 | 27.02 | 35.68 |
| 3D Box | SegVol NORM | 3 | 0.55 | 40.67 | 68.11 | 69.72 | 63.21 | 50.13 | 89.95 | 79.79 | 57.77 |

Table 4: Experimental results simulating a realistic clinician's effort. 'PPV' stands for Point Per Volume, that was sampled from the center of the target object.

## C.3 ITERATIVE REFINEMENT 2D

| Iteration | Prompter | Model | Interactions | D1 | D2 | D3 | D4 | D5 | D6 | D7 | D8 | Average |
|---|---|---|---|---|---|---|---|---|---|---|---|---|
| 0 | 1PPS + 1PPS Refine | SAM | 1X/1X | 0.81 | 7.63 | 3.13 | 2.06 | 1.75 | 1.00 | 7.86 | 12.47 | 4.59 |
| 1 | 1PPS + 1PPS Refine | SAM | 1X/1X | 1.02 | 7.91 | 4.46 | 3.22 | 2.64 | 1.13 | 8.75 | 13.81 | 5.37 |
| 2 | 1PPS + 1PPS Refine | SAM | 1X/1X | 1.11 | 7.94 | 4.92 | 3.66 | 3.30 | 1.41 | 9.91 | 15.54 | 5.97 |
| 3 | 1PPS + 1PPS Refine | SAM | 1X/1X | 1.18 | 7.36 | 5.26 | 4.42 | 4.04 | 1.55 | 11.04 | 16.95 | 6.47 |
| 4 | 1PPS + 1PPS Refine | SAM | 1X/1X | 1.18 | 7.05 | 4.64 | 4.95 | 4.21 | 1.62 | 12.06 | 17.69 | 6.68 |
| 5 | 1PPS + 1PPS Refine | SAM | 1X/1X | 1.12 | 7.20 | 3.31 | 5.55 | 4.37 | 1.42 | 12.53 | 18.23 | 6.72 |
| 0 | 1PPS + 1PPS Refine | SAM2 | 1X/1X | 1.25 | 9.31 | 4.08 | 3.90 | 3.54 | 1.51 | 9.60 | 16.35 | 6.19 |
| 1 | 1PPS + 1PPS Refine | SAM2 | 1X/1X | 2.96 | 11.29 | 8.69 | 6.20 | 6.03 | 3.32 | 18.10 | 24.31 | 10.11 |
| 2 | 1PPS + 1PPS Refine | SAM2 | 1X/1X | 5.25 | 12.89 | 11.67 | 8.13 | 8.15 | 5.05 | 24.99 | 29.54 | 13.21 |
| 3 | 1PPS + 1PPS Refine | SAM2 | 1X/1X | 7.52 | 14.56 | 15.01 | 10.19 | 10.40 | 6.87 | 30.15 | 33.57 | 16.03 |
| 4 | 1PPS + 1PPS Refine | SAM2 | 1X/1X | 9.81 | 16.22 | 18.34 | 12.17 | 12.30 | 8.26 | 33.94 | 48.51 | 19.94 |
| 5 | 1PPS + 1PPS Refine | SAM2 | 1X/1X | 12.14 | 17.78 | 21.41 | 13.73 | 14.36 | 9.20 | 37.63 | 51.30 | 22.19 |
| 0 | 1PPS + 1PPS Refine | SamMed 2D | 1X/1X | 10.72 | 32.14 | 24.55 | 28.83 | 24.42 | 9.71 | 30.83 | 31.24 | 24.06 |
| 1 | 1PPS + 1PPS Refine | SamMed 2D | 1X/1X | 12.44 | 36.15 | 28.54 | 31.35 | 25.82 | 11.78 | 39.24 | 37.54 | 27.86 |
| 2 | 1PPS + 1PPS Refine | SamMed 2D | 1X/1X | 12.49 | 37.54 | 30.25 | 32.28 | 26.04 | 12.41 | 43.73 | 40.46 | 29.40 |
| 3 | 1PPS + 1PPS Refine | SamMed 2D | 1X/1X | 12.59 | 38.43 | 31.49 | 32.85 | 26.31 | 12.88 | 46.71 | 42.32 | 30.45 |
| 4 | 1PPS + 1PPS Refine | SamMed 2D | 1X/1X | 12.78 | 39.12 | 32.60 | 33.31 | 26.63 | 13.31 | 48.91 | 43.68 | 31.29 |
| 5 | 1PPS + 1PPS Refine | SamMed 2D | 1X/1X | 12.93 | 39.70 | 33.67 | 33.74 | 27.00 | 13.67 | 50.53 | 44.81 | 32.01 |
| 0 | 1PPS + Scribble Refine | SAM | 1/3 | 0.81 | 7.63 | 3.13 | 2.06 | 1.53 | 1.00 | 7.86 | 12.47 | 4.56 |
| 1 | 1PPS + Scribble Refine | SAM | 1/3 | 0.89 | 7.67 | 4.53 | 3.24 | 2.21 | 1.18 | 7.50 | 11.65 | 4.86 |
| 2 | 1PPS + Scribble Refine | SAM | 1/3 | 0.93 | 7.75 | 4.45 | 3.54 | 2.34 | 1.37 | 7.96 | 11.24 | 4.95 |
| 3 | 1PPS + Scribble Refine | SAM | 1/3 | 0.96 | 7.15 | 3.50 | 3.49 | 2.36 | 1.40 | 7.94 | 10.60 | 4.67 |
| 4 | 1PPS + Scribble Refine | SAM | 1/3 | 0.92 | 6.97 | 2.08 | 3.43 | 2.07 | 1.40 | 7.97 | 9.84 | 4.33 |
| 5 | 1PPS + Scribble Refine | SAM | 1/3 | 0.88 | 6.75 | 1.41 | 3.43 | 1.93 | 1.14 | 7.59 | 9.53 | 4.08 |
| 0 | 1PPS + Scribble Refine | SAM2 | 1/3 | 1.25 | 8.82 | 4.08 | 3.90 | 3.51 | 1.51 | 9.60 | 16.35 | 6.13 |
| 1 | 1PPS + Scribble Refine | SAM2 | 1/3 | 1.74 | 10.52 | 8.13 | 5.39 | 6.26 | 3.01 | 16.18 | 18.28 | 8.69 |
| 2 | 1PPS + Scribble Refine | SAM2 | 1/3 | 2.18 | 11.02 | 10.05 | 5.84 | 7.91 | 4.14 | 19.20 | 19.15 | 9.94 |
| 3 | 1PPS + Scribble Refine | SAM2 | 1/3 | 2.52 | 11.37 | 10.93 | 6.16 | 8.66 | 5.06 | 21.45 | 25.97 | 11.52 |
| 4 | 1PPS + Scribble Refine | SAM2 | 1/3 | 2.83 | 11.56 | 12.37 | 6.26 | 9.23 | 5.88 | 24.95 | 27.84 | 12.61 |
| 5 | 1PPS + Scribble Refine | SAM2 | 1/3 | 3.10 | 11.90 | 13.38 | 6.26 | 9.50 | 6.93 | 26.28 | 30.02 | 13.42 |
| 0 | 1PPS + Scribble Refine | SamMed 2D | 1/3 | 10.72 | 32.14 | 24.55 | 28.83 | 23.56 | 9.71 | 30.83 | 31.24 | 23.95 |
| 1 | 1PPS + Scribble Refine | SamMed 2D | 1/3 | 11.80 | 34.79 | 26.82 | 29.72 | 23.73 | 11.32 | 37.22 | 36.70 | 26.51 |
| 2 | 1PPS + Scribble Refine | SamMed 2D | 1/3 | 11.71 | 35.29 | 27.88 | 29.50 | 23.33 | 11.63 | 40.64 | 38.77 | 27.34 |
| 3 | 1PPS + Scribble Refine | SamMed 2D | 1/3 | 11.55 | 35.68 | 28.26 | 29.15 | 23.23 | 11.68 | 42.57 | 39.92 | 27.75 |
| 4 | 1PPS + Scribble Refine | SamMed 2D | 1/3 | 11.49 | 35.87 | 28.33 | 29.19 | 23.08 | 11.76 | 43.94 | 40.71 | 28.05 |
| 5 | 1PPS + Scribble Refine | SamMed 2D | 1/3 | 11.46 | 36.05 | 28.72 | 28.87 | 22.97 | 11.88 | 44.94 | 41.32 | 28.28 |
| 0 | 3B Inter + Scribble Refine | SAM | 6/3 | 13.13 | 56.31 | 63.34 | 58.64 | 65.01 | 71.35 | 65.52 | 61.78 | 56.89 |
| 1 | 3B Inter + Scribble Refine | SAM | 6/3 | 0.99 | 8.66 | 3.25 | 3.53 | 2.88 | 6.14 | 8.04 | 11.86 | 5.67 |
| 2 | 3B Inter + Scribble Refine | SAM | 6/3 | 0.85 | 7.03 | 1.29 | 3.17 | 9.95 | 5.28 | 6.89 | 14.31 | 6.10 |
| 3 | 3B Inter + Scribble Refine | SAM | 6/3 | 0.80 | 9.06 | 2.94 | 2.19 | 3.49 | 3.36 | 7.01 | 9.84 | 4.84 |
| 4 | 3B Inter + Scribble Refine | SAM | 6/3 | 0.71 | 7.49 | 2.01 | 2.21 | 2.09 | 1.89 | 6.12 | 9.09 | 3.95 |
| 5 | 3B Inter + Scribble Refine | SAM | 6/3 | 0.76 | 7.03 | 2.46 | 2.52 | 2.33 | 1.42 | 5.98 | 9.38 | 3.99 |
| 0 | 3B Inter + Scribble Refine | SAM2 | 6/3 | 69.76 | 63.85 | 67.71 | 69.08 | 69.47 | 73.11 | 68.36 | 63.24 | 68.07 |
| 1 | 3B Inter + Scribble Refine | SAM2 | 6/3 | 10.03 | 27.28 | 25.37 | 19.56 | 24.35 | 18.26 | 23.87 | 27.85 | 22.07 |
| 2 | 3B Inter + Scribble Refine | SAM2 | 6/3 | 5.03 | 12.93 | 10.53 | 7.01 | 9.51 | 3.99 | 19.74 | 19.59 | 11.04 |
| 3 | 3B Inter + Scribble Refine | SAM2 | 6/3 | 4.21 | 10.53 | 7.80 | 6.68 | 6.67 | 3.72 | 15.48 | 16.39 | 8.93 |
| 4 | 3B Inter + Scribble Refine | SAM2 | 6/3 | 3.97 | 10.09 | 6.96 | 7.03 | 5.43 | 3.54 | 14.44 | 14.94 | 8.30 |
| 5 | 3B Inter + Scribble Refine | SAM2 | 6/3 | 3.38 | 9.99 | 6.69 | 5.64 | 4.26 | 2.80 | 13.58 | 13.44 | 7.47 |
| 0 | 3B Inter + Scribble Refine | SamMed 2D | 6/3 | 27.83 | 48.08 | 52.70 | 60.66 | 46.19 | 38.63 | 55.31 | 55.81 | 48.15 |
| 1 | 3B Inter + Scribble Refine | SamMed 2D | 6/3 | 28.40 | 51.32 | 57.53 | 63.33 | 45.75 | 41.42 | 61.61 | 59.73 | 51.14 |
| 2 | 3B Inter + Scribble Refine | SamMed 2D | 6/3 | 24.29 | 49.42 | 53.02 | 59.24 | 40.66 | 35.69 | 62.87 | 58.85 | 48.00 |
| 3 | 3B Inter + Scribble Refine | SamMed 2D | 6/3 | 20.91 | 46.94 | 47.48 | 52.73 | 35.76 | 28.20 | 62.12 | 54.89 | 43.63 |
| 4 | 3B Inter + Scribble Refine | SamMed 2D | 6/3 | 18.76 | 44.63 | 41.46 | 46.50 | 32.03 | 23.98 | 58.80 | 51.86 | 39.75 |
| 5 | 3B Inter + Scribble Refine | SamMed 2D | 6/3 | 17.07 | 42.27 | 37.39 | 41.62 | 29.73 | 20.80 | 54.00 | 49.27 | 36.52 |
| 0 | 3P Inter + Scribble Refine | SAM | 5/3 | 0.84 | 8.45 | 3.50 | 2.17 | 1.77 | 1.02 | 8.29 | 12.61 | 4.83 |
| 1 | 3P Inter + Scribble Refine | SAM | 5/3 | 0.89 | 8.26 | 4.59 | 3.58 | 2.54 | 1.21 | 7.96 | 11.68 | 5.09 |
| 2 | 3P Inter + Scribble Refine | SAM | 5/3 | 0.98 | 8.15 | 5.18 | 4.21 | 2.73 | 1.44 | 8.62 | 11.54 | 5.36 |
| 3 | 3P Inter + Scribble Refine | SAM | 5/3 | 1.03 | 7.50 | 3.61 | 4.13 | 2.51 | 1.50 | 9.18 | 10.95 | 5.05 |
| 4 | 3P Inter + Scribble Refine | SAM | 5/3 | 1.03 | 7.07 | 2.21 | 4.14 | 2.44 | 1.48 | 8.84 | 10.41 | 4.70 |
| 5 | 3P Inter + Scribble Refine | SAM | 5/3 | 0.99 | 6.93 | 1.49 | 4.16 | 1.99 | 1.32 | 8.19 | 9.76 | 4.35 |
| 0 | 3P Inter + Scribble Refine | SAM2 | 5/3 | 1.38 | 8.67 | 4.73 | 4.10 | 4.21 | 1.63 | 10.46 | 16.98 | 6.52 |
| 1 | 3P Inter + Scribble Refine | SAM2 | 5/3 | 1.87 | 10.11 | 8.77 | 6.04 | 8.03 | 3.47 | 16.69 | 19.63 | 9.33 |
| 2 | 3P Inter + Scribble Refine | SAM2 | 5/3 | 2.37 | 10.72 | 11.33 | 7.02 | 10.49 | 4.62 | 22.04 | 21.17 | 11.22 |
| 3 | 3P Inter + Scribble Refine | SAM2 | 5/3 | 2.84 | 11.10 | 13.65 | 7.62 | 11.69 | 5.90 | 23.87 | 27.73 | 13.05 |
| 4 | 3P Inter + Scribble Refine | SAM2 | 5/3 | 3.33 | 11.32 | 14.69 | 7.84 | 12.80 | 6.96 | 27.67 | 28.86 | 14.18 |
| 5 | 3P Inter + Scribble Refine | SAM2 | 5/3 | 3.50 | 11.52 | 15.77 | 7.97 | 13.55 | 7.65 | 30.59 | 30.82 | 15.17 |
| 0 | 3P Inter + Scribble Refine | SamMed 2D | 5/3 | 11.71 | 33.07 | 28.46 | 34.42 | 28.34 | 10.88 | 36.81 | 35.65 | 27.42 |
| 1 | 3P Inter + Scribble Refine | SamMed 2D | 5/3 | 12.72 | 34.74 | 30.27 | 34.05 | 27.97 | 12.06 | 40.92 | 38.13 | 28.86 |
| 2 | 3P Inter + Scribble Refine | SamMed 2D | 5/3 | 12.46 | 35.12 | 30.58 | 33.13 | 27.38 | 12.19 | 42.79 | 39.25 | 29.11 |
| 3 | 3P Inter + Scribble Refine | SamMed 2D | 5/3 | 12.25 | 35.25 | 30.53 | 32.57 | 26.95 | 12.28 | 43.89 | 39.98 | 29.21 |
| 4 | 3P Inter + Scribble Refine | SamMed 2D | 5/3 | 12.13 | 35.30 | 30.42 | 32.12 | 26.67 | 12.33 | 44.87 | 40.52 | 29.30 |
| 5 | 3P Inter + Scribble Refine | SamMed 2D | 5/3 | 12.06 | 35.34 | 30.42 | 31.66 | 26.43 | 12.41 | 45.82 | 40.97 | 29.39 |

Table 5: Interactive refinement results for 2D models across 5 iterations. The initial prediction is made either using a single point per slice or one of our proposed prompting schemes. Omitting the previous point during refinement led to a drop in performance. We compared the unrealistic slice-wise refinement (1 interaction per slice) our proposed scribble refinement method (3 interactions)

## C.4 ITERATIVE REFINEMENT 3D

| Iteration | Prompter | Model | Interactions | D1 | D2 | D3 | D4 | D5 | D6 | D7 | D8 | Average |
|---|---|---|---|---|---|---|---|---|---|---|---|---|
| 0 | 1 PPV + 1 PPV Refine | SamMed 3D | 1/1 | 2.04 | 17.01 | 24.06 | 27.16 | 15.09 | 19.64 | 72.66 | 53.04 | 28.84 |
| 1 | 1 PPV + 1 PPV Refine | SamMed 3D | 1/1 | 3.20 | 17.85 | 27.58 | 36.18 | 14.08 | 23.34 | 73.30 | 79.56 | 34.39 |
| 2 | 1 PPV + 1 PPV Refine | SamMed 3D | 1/1 | 4.11 | 17.75 | 28.43 | 35.23 | 12.62 | 23.84 | 72.95 | 83.39 | 34.79 |
| 3 | 1 PPV + 1 PPV Refine | SamMed 3D | 1/1 | 4.54 | 17.74 | 27.94 | 37.40 | 11.32 | 24.33 | 72.22 | 84.95 | 35.06 |
| 4 | 1 PPV + 1 PPV Refine | SamMed 3D | 1/1 | 4.81 | 17.78 | 27.61 | 34.85 | 10.64 | 24.73 | 72.74 | 85.16 | 34.79 |
| 5 | 1 PPV + 1 PPV Refine | SamMed 3D | 1/1 | 4.94 | 17.71 | 27.35 | 35.67 | 10.27 | 24.70 | 71.98 | 85.14 | 34.72 |
| 0 | 1 PPV + 1 PPV Refine | SamMed 3D Turbo | 1/1 | 5.12 | 31.28 | 46.07 | 34.33 | NaN | 46.38 | 82.95 | 59.37 | 43.64 |
| 1 | 1 PPV + 1 PPV Refine | SamMed 3D Turbo | 1/1 | 5.65 | 32.69 | 48.04 | 37.87 | NaN | 50.14 | 86.45 | 71.83 | 47.53 |
| 2 | 1 PPV + 1 PPV Refine | SamMed 3D Turbo | 1/1 | 5.73 | 33.18 | 48.71 | 43.93 | NaN | 51.80 | 87.17 | 78.20 | 49.82 |
| 3 | 1 PPV + 1 PPV Refine | SamMed 3D Turbo | 1/1 | 5.76 | 33.69 | 48.48 | 47.16 | NaN | 52.98 | 87.61 | 80.47 | 50.88 |
| 4 | 1 PPV + 1 PPV Refine | SamMed 3D Turbo | 1/1 | 5.91 | 34.24 | 48.81 | 48.17 | NaN | 54.06 | 87.63 | 81.89 | 51.53 |
| 5 | 1 PPV + 1 PPV Refine | SamMed 3D Turbo | 1/1 | 6.22 | 34.51 | 49.62 | 50.09 | NaN | 54.35 | 88.01 | 83.72 | 52.36 |
| 0 | 1 PPV + Scribble Refine | SamMed 3D | 1/3 | 2.04 | 17.01 | 24.06 | 27.16 | 15.09 | 19.64 | 72.66 | 53.04 | 28.84 |
| 1 | 1 PPV + Scribble Refine | SamMed 3D | 1/3 | 3.25 | 17.38 | 25.15 | 32.79 | 13.53 | 23.44 | 71.94 | 68.83 | 32.04 |
| 2 | 1 PPV + Scribble Refine | SamMed 3D | 1/3 | 3.98 | 17.69 | 25.86 | 35.22 | 11.85 | 24.94 | 72.82 | 77.12 | 33.69 |
| 3 | 1 PPV + Scribble Refine | SamMed 3D | 1/3 | 4.43 | 17.66 | 25.95 | 36.58 | 11.02 | 25.13 | 73.30 | 82.72 | 34.60 |
| 4 | 1 PPV + Scribble Refine | SamMed 3D | 1/3 | 4.57 | 17.78 | 26.47 | 36.33 | 10.26 | 25.43 | 73.30 | 84.00 | 34.77 |
| 5 | 1 PPV + Scribble Refine | SamMed 3D | 1/3 | 4.75 | 18.01 | 26.81 | 37.68 | NaN | 25.80 | 72.88 | 86.39 | 38.90 |
| 0 | 1 PPV + Scribble Refine | SamMed 3D Turbo | 1/3 | 5.12 | 31.28 | 46.07 | 34.33 | NaN | 46.38 | 82.95 | 59.37 | 43.64 |
| 1 | 1 PPV + Scribble Refine | SamMed 3D Turbo | 1/3 | 5.44 | 31.59 | 47.68 | 37.93 | NaN | 48.96 | 86.40 | 70.52 | 46.93 |
| 2 | 1 PPV + Scribble Refine | SamMed 3D Turbo | 1/3 | 4.88 | 32.01 | 47.72 | 40.62 | NaN | 50.76 | 87.19 | 76.26 | 48.49 |
| 3 | 1 PPV + Scribble Refine | SamMed 3D Turbo | 1/3 | 4.42 | 32.55 | 47.42 | 41.96 | NaN | 52.29 | 87.73 | 78.65 | 49.29 |
| 4 | 1 PPV + Scribble Refine | SamMed 3D Turbo | 1/3 | 4.32 | 33.15 | 48.10 | 43.95 | NaN | 53.40 | 88.11 | 80.05 | 50.16 |
| 5 | 1 PPV + Scribble Refine | SamMed 3D Turbo | 1/3 | 4.35 | 33.58 | 49.16 | 45.71 | NaN | 54.56 | 88.30 | 80.86 | 50.93 |

Table 6: Interactive refinement results for 3D models over 5 iterations. The initial interaction always starts from a central point within the target class, and refinement is performed either by randomly sampling positive or negative points (1 interaction) or by selecting a point using the proposed scribble refinement method. Scribble drawing is counted as three interactions. Including the previous point produced worse for 3D models.

