# OpenReview forum: "INTRABENCH: Interactive Radiological Benchmark"
_ICLR.cc/2025/Conference — ICLR 2025 Conference Withdrawn Submission_

### Official Review · Reviewer_9NVy · 2024-10-31

**Soundness:** 1
**Presentation:** 2
**Contribution:** 2
**Rating:** 1
**Confidence:** 5

**Summary:**

Aiming to find the best interactive segmentation methods, this benchmarking study compares seven SAM-based models across eight datasets, encompassing various target structures and imaging modalities.

**Strengths:**

This study benchmarks seven existing SAM-based interactive segmentation methods and provides an extendable framework to incorporate new methods and prompting strategies. A fair comparison is crucial and commendable, as these similar methods all claim superiority yet differ significantly in training data, architectures, and prompts, making it challenging to identify the most suitable approach for a specific application.

**Weaknesses:**

1. The eight benchmarking datasets do not include 2D medical data, such as X-rays or pathology images. As a result, the suitability and effectiveness of each method for 2D images remain unclear. While 3D models may not be suitable for segmenting 2D images, separate comparisons on 2D and 3D datasets would provide valuable insights. Additionally, the benchmarking datasets are not large and diverse enough.
2. Degrees of freedom (DoF) do not objectively reflect human effort, making DoF an unreliable measure of effort. For example, a 2D box has 5 DoF, while a 3D box has 6 DoF. However, the difficulty of drawing a 3D box is substantially more than 1.2 times that of a 2D box.
3. Point interpolation may be unsuitable for certain shapes, such as arch-shaped structures, as the interpolated points can fall outside the object of interest.
4. I doubt the suitability of this benchmarking paper for ICML, given its lack of technical novelty and limited interesting takeaways.

**Questions:**

1. The study is limited to SAM-based interactive segmentation methods. Why were non-SAM-based methods not considered? It is important to clearly give?
2. In lines 139-140, it’s mentioned that IntRaBench was created to address the pitfalls outlined in lines 84-132. However, it’s unclear why addressing these issues should be the goal of a benchmarking paper. Is the proposed framework able to address these issues simultaneously? Or was any new method proposed in the paper? Correct me if I missed it.
3. The SAM-based methods compared in this paper were trained on different datasets and used out-of-the-box, raising questions about the fairness of the comparison given the variability in training data. How do the authors address fairness when comparing these diverse methods?
4. What is the authors’ perspective on using degrees of freedom as a measure of human effort?

---

### Official Review · Reviewer_KRPa · 2024-11-01

**Soundness:** 3
**Presentation:** 2
**Contribution:** 3
**Rating:** 3
**Confidence:** 4

**Summary:**

The authors propose IntRaBench, a framework to ensure fair evaluation of interactive segmentation methods. It addresses limitations of current approaches by providing a realistic, clinically relevant framework that includes diverse datasets, target structures, and segmentation models.

**Strengths:**

The benchmark offers a flexible codebase for integrating new models and prompting strategies, ensuring fair comparisons between 2D and 3D models. The authors demonstrate the effectiveness of IntRaBench through experiments that highlight the importance of bounding boxes over points and the necessity of iterative refinement for accurate segmentation.

**Weaknesses:**

1. While the paper simulates clinical settings, it lacks real-world clinical validation involving actual clinicians. The authors should include studies to assess the prompting strategies in actual clinical workflows. e.g. How to efficient utilize these methods to reduce the manual segmentation time cost in clinical scenarios? Besides, the evaluation are conducted on relatively small datasets with limited modalities (only CT and MRI). It would be better to conduct experiments on large-scale datasets (like AbdomenAtlas [1] and FLARE 24 [2] with 10000+ volumes) and more modalities like PET,  Ultrasound, etc.

[1] AbdomenAtlas: A large-scale, detailed-annotated, & multi-center dataset for efficient transfer learning and open algorithmic benchmarking, MedIA 2024.
[2] https://www.codabench.org/competitions/2296/

2. What is the detail of the "framework"? An out-of-the-box segmentation tool without coding? Besides, It would better to visualize with some real cases to help readers better understand the workflow. Currently, it seems only a re-implementation of existing SAM-based methods under the same settings.

3. Figures need to be redrawn as their resolution is too poor.

4. Other than DSC, it would be better to add more evaluation metrics. e.g. boundary-based NSD. Besides, the computation efficiency should also be compared.



Typo: line 493 ??

**Questions:**

My major concern is the limited novelty of the work. As a benchmark study, the experiments are only conducted on relative small datasets with limited modalities. Several conclusions drawn in the paper seem to align with those found in prior studies [1,2]. The so-called realistic, clinically relevant framework lacks of real clinical analysis and clear visualization.

[1] Segment anything model for medical image analysis: an experimental study. MedIA 2023.
[2] Segment anything model for medical images? MedIA 2024.

---

### Official Review · Reviewer_oZ5Z · 2024-11-04

**Soundness:** 3
**Presentation:** 2
**Contribution:** 2
**Rating:** 5
**Confidence:** 4

**Summary:**

The paper evaluates various open set segmentation models on the iterative segmentation task. This includes both 2D and 3D Segmentation models and points vs bbox prompts. To evaluate 2D models on realistic human efforts rather than the unrealistic slice-by-slice segmentation, initial prompts and iterative scribble-based prompts are interpolated or propagated, rather than slice-by-slice prompts.

These interactive open-set methods were tested on recent MRI and CT datasets (8 datasets in total) segmenting mostly pathological structures (lesions and tumors).

Codebase is provided for reproducibility and future extension which includes 8 model implementations, scripts for data ingestion/preprocessing, framework for point- and bounding box-based automatic iterative prompting for refining the initial segmentation.

Based on the framework, the superiority of bounding-box based refinement compared to point-based is shown as well as the gap in performance for deep-learning based segmentation of pathological structure in medical 3D volume is shown.

**Strengths:**

The paper points out disparities and lack of details in evaluating 2D and 3D open-set foundation segmentation models.  Case in Point: Lack of details required for reproducibility, such as whether a 3-point prompt was provided iteratively or all at the same time.

Especially, the issue of fairly evaluating 2D and 3D iterative segmentation models based on realistic human effort has been well executed via point/box propagation of 2D prompts from 2D to 3D volume.

The paper is well written for the most part with major ideas clearly explained. Figures (for example, Figure 3) really help get the major ideas across to the readers.

The code base and the framework are made available, extendible and reproducible, including data preprocessing scripts. The abstraction and Object-oriented programming paradigm used in the code base makes the code easy to follow and hopefully easy to extend.

**Weaknesses:**

The benchmarking framework and evaluation of the 2D and 3D open set segmentation models is useful, in and of itself. But having done the heavy lifting of bringing various segmentation models and evaluation framework for iterative segmentation within a same framework, additional experiments could strengthen the usefulness of such an evaluation framework.  This is especially in light to the similar results/conclusions in earlier papers, e.g. segvol paper, (also cited by the proposed paper) that show bbox's superiority over point prompts and the gap in performance for healthy organs vs pathological ones (85% DSC for AMOS22 vs 71% for ULS23). The proposed paper has potential to go beyond these results and provide comprehensive benchmarking of, especially, the iterative prompting strategies, for example.

For example, additional experiments such as:
- Difference in performance between center clicks (obtaining maximum information per interaction) vs other (may be less informative or random or border) clicks
 - Difference in performance between various scribble strategies. For example, say between the one presented in the paper vs. The Scribble-prompt paper
 - Performance disaggregation for healthy organs vs pathological structures

The code base could use more documentation and test case coverage. Additionally, the writing could use more polishing. Details below.

**Questions:**

- Could you please provide a rationale for using different scribble generation strategies for false positives and false negatives? Could the same strategy be used for both? Also, did you explore other scribble generation strategies, say example, scribble-prompt [Wong et.al 2023]? Additionally, it helps the reader if a visual example of the scribble generation for the negative prompts is provided.

- Table 5 is available in the supplementary section but is referred to in the main text. May be shortening section 2.3, for example, keeping the gist intact and bringing useful results from supplementary to the results section in a more succinct visual form might strengthen the paper.
The writing could use some polishing. There are broken sentences requiring clean up (line 406 “Denote the different y-axis scalings”), typos, etc.

- Typos : Line 40 citation format, Line 45 inverted comma ”SAM” -> \“SAM”, Line 106 ’too small’ -> \‘too small’, Sec3.3 Bullet Point 3??, Figure 3 caption: promoting -> prompting, Line 356 sparsitity -> sparsity

- The title could be more specific suggesting the benchmark is focused on segmentation task on 3D Medical Images with interactive/iterative refinement rather than the current generic radiology benchmarking.

- In figure 4, to infer that 2D bboxes are better than point prompts, the reader has to take information from two different plots. Would it make more sense to merge the two plots, say for example, have a single line plot or bar plot for both cases?

- Can you please cite the “previous literature” that is mentioned in line 496?

- The code base could use more documentation and test case coverage.

---

### Note · Authors · 2024-11-13

**Comment:**

Given the Feedback of the reviewers we withdraw this paper in it's current form from ICLR.
While valid criticisms about deeper insights have been raised by Reviewer oZ5Z, we want to point out that Reviewers KRPa and especially 9NVy seem to have missed the point of the benchmark. To avoid this in future versions we will make sure to communicate this even more clearly in the manuscripts:

Criticisms that were raised that we deem unreasonable and show a lack of understanding:

>The benchmark being evaluated on too small datasets instead of FLARE24 or Abdomen Atlas

In the paper we clearly state that these datasets are not useable as __everyone__ trained on them, hence introducing train-test leakage if we use them and they are a composition of other datasets, increasing the danger of train-test leakage. Moreover, we clearly state that the focus of the benchmark is on pathologies, as supervised models can easily handle organ segmentation (with better performance). Hence all pathological/lesion dataset sizes are naturally smaller as pathologies are rarer. We will make sure to clarify this even more clearly in a revised version, but we believe it is already apparent from the manuscript.

>The benchmark only evaluates 3D data and not 2D histopatho or x-rays

Well yes, this is the entire point of the benchmark. Previous benchmarks only evaluated in 2D fashion, while this benchmark is about a realistic evaluation of 2D capable methods and native 3D methods in the __3D domain__.

> Benchmark is not tested in a clinically realistic setting

We clearly state this is a current limitation of the paper, but our proposed proxy for "human effort" is a first step in this direction.

>  Degrees of freedom (DoF) do not objectively reflect human effort

Yes, we explain so in the paper L282ff and don't use it because of this.

 > The study is limited to SAM-based interactive segmentation methods. Why were non-SAM-based methods not considered? It is important to clearly give?

This is a false statement, we use __all__ open-set prompt-based segmentation methods commonly known. If there should be some that we missed, it would have been great if the names and references of them would have been stated, so we can improve on it. Other methods are not included because none tick these boxes and applying a close-set model to an open-set problem would be unfair (as we specified in the manuscript already).

> I doubt the suitability of this benchmarking paper for ICML

It's a review for ICLR....

Overall we want to thank Reviewer oZ5Z and KRPa for constructive criticism that allowed us to improve our paper, while we want to express our disappointment in Reviewer 9NVy's comments. The Reviewer clearly has not read the paper thoroughly hence the majority of points raised are clearly missing the mark.

**Withdrawal Confirmation:**

I have read and agree with the venue's withdrawal policy on behalf of myself and my co-authors.